ecology, environmental science

seaweed, historical ecology, herbaria, stable isotopes, upwelling, ocean memory

**Author for correspondence:**
Emily A. Miller
e-mail: emiller@mbari.org

# Herbaria macroalgae as a proxy for historical upwelling trends in Central California

Emily A. Miller[1], Susan E. Lisin[1], Celia M. Smith[2] and Kyle S. Van Houtan[1,3]

[1]Conservation Research Department, Monterey Bay Aquarium, 886 Cannery Row, Monterey, CA 93950, USA
[2]Botany Department, University of Hawai'i at Manoa, 3190 Maile Way, Honolulu, Hawai'i 96822, USA
[3]Nicholas School of the Environment, Duke University, Box 90328, Durham, NC 27708, USA

EAM, 0000-0002-3281-571X; KSVH, 0000-0001-5725-1773

Planning for future ocean conditions requires historical data to establish more informed ecological baselines. To date, this process has been largely limited to instrument records and observations that begin around 1950. Here, we show how marine macroalgae specimens from herbaria repositories may document long-term ecosystem processes and extend historical information records into the nineteenth century. We tested the effect of drying and pressing six macroalgae species on amino acid, heavy metal and bulk stable isotope values over 1 year using modern and archived paper. We found historical paper composition did not consistently affect values. Certain species, however, had higher variability in particular metrics while others were more consistent. Multiple herbaria provided *Gelidium* (Rhodophyta) samples collected in southern Monterey Bay from 1878 to 2018. We examined environmental relationships and found $\delta^{15}N$ correlated with the Bakun upwelling index, the productivity regime of this ecosystem, from 1946 to 2018. Then, we hindcasted the Bakun index using its derived relationship with *Gelidium* $\delta^{15}N$ from 1878 to 1945. This hindcast provided new information, observing an upwelling decrease mid-century leading up to the well-known sardine fishery crash. Our case study suggests marine macroalgae from herbaria are an underused resource of the marine environment that precedes modern scientific data streams.

## 1. Introduction

Long-term environmental records are critical for understanding natural variability, establishing informed ecological baselines and designing effective management strategies [1,2]. Coastal environments have seen dramatic ecological changes over the last 150 years [3–5]; however, much of this period is scientifically undocumented. Marine organisms may fill important data gaps and extend time series, as numerous and diverse taxa archive environmental conditions within their tissues [6–11], recording an 'ocean memory' [12]. As a result, natural history repositories of marine algae, foraminifera, corals, birds, reptiles and other taxa may facilitate reconstruction of historical environmental conditions to complement and extend instrument records.

Specimens must be temporally and spatially resolved to reconstruct oceanographic or ecological conditions. Calcareous species such as corals and rhodoliths have been used to record changing ocean conditions through time by taking multiple measurements from the same organism with known chronological structure [7,13]. Multiple individuals from the same species can also be sampled over time to record ocean trophic dynamics and environmental change [8]. Marine macroalgae have been used to document the effect of anthropogenic pollutants in coastal ecosystems [14,15], and more recently have been used to document long-term change of local marine community structure [16].

Macroalgae herbarium collections provide long-term ocean records from the nineteenth century in most of Europe and North America or even centuries earlier in Germany and Italy [17]. In the Victorian era, in western Europe and North America, scientists as well as the middle class public became fascinated by pressing seaweeds [18]. People began collecting intricate and diverse species for artistic personal collections and scientific institutional collections. Many of these herbaria are still present and active today, though they are underused in research and continued systematic collecting for regional floras is underfunded [19]. While we have instrumentation to measure environmental change into the future, traditionally processed collections [20] are necessary to track impacts to species and ecosystems through time [21,22]. Before researchers can access these environmental proxies from herbarium collections, we need to know that the archived specimens are reliable. The chemical makeup of each stored specimen must be representative of that of the organism as it experienced the ocean.

Here, we examine whether herbaria specimens reliably represent historical ocean conditions using a methodological experiment and a case study of historical herbaria specimens. First, we determine the effect of conventional drying and pressing marine macroalgae over 1 year on focal tissue components: amino acid and protein composition, heavy metal concentrations and stable isotope ratios. Macroalgae were historically pressed on paper containing materials not found in today's modern, acid-free herbarium paper. To examine any differences between the two papers that may require consideration when interpreting older specimens, we press each species on both modern and archived paper. We resample the same individuals during a curing period to record any changes associated with drying time or paper type. Second, given these methodological experiments, we use archived specimens of one focal genus from multiple herbarium collections to examine changing coastal ocean conditions over 140 years in Monterey Bay, California. This method may not only extend upwelling indices for other eastern boundary current systems (e.g. Canary, Humboldt and Benguela), but may broadly inform ecological indices relevant for marine coastal management.

## 2. Methods

### (a) Modern and historical algae samples

To begin, we selected six algae species that are native to the region, represent morphological and taxonomic diversity, and are present throughout the last 140 years in local herbaria. We selected three blade/sheet structured species and three branching species that fit these criteria—*Cryptopleura ruprechtiana* (Rhodophyta), *Ulva californica* (Chlorophyta) and *Macrocystis pyrifera* (Phaeophyceae, Ochrophyta); and *Gelidium pupurascens* (Rhodophyta), *Cladophora columbiana* (Chlorophyta) and *Silvetia compressa* (Phaeophyceae, Ochrophyta), respectively.

We collected specimens for all six species from Cabrillo Point at Hopkins Marine Station, Pacific Grove, California (36.621° N, 121.903° W) in June 2018 (electronic supplementary material, figure S1). We rinsed specimens in filtered seawater, reserving one sixth from each species collection to oven-dry at 55°C for 72 h. We prepared the remaining tissue in traditional botanical lattice press frames [23] with half of the material on archived herbarium paper (circa 1930) and the remainder on pH neutral cotton fibre paper (Herbarium Supply Co., University of California Type mounting no. 102). To expedite desiccation and avoid mould, we regularly changed the blotting paper in the pressing (Herbarium Supply Co., heavy white driers no. 221) until all specimens were dry (<21 days). After pressing, we transferred the specimens to herbarium folders (Herbarium Supply Co., Missouri Botanical Garden genus covers no. 135) and stored the folders in museum cabinets (Steel Fixture Manufacturing, model SS4824) at the Ocean Memory Laboratory in the Monterey Bay Aquarium.

Owing to its consistent representation in the herbaria (1878–2018) and performance in the curing experiment (electronic supplementary material, figures S2–S5), we used *Gelidium* spp. for long-term reconstructions. *Gelidium* spp. are also favourable as their branching morphology minimizes aesthetic impacts from partial specimen sampling. We sourced $n = 70$ *Gelidium* samples from specimens at the Gilbert M. Smith Herbarium at the Hopkins Marine Station ($n = 39$), Monterey Bay Aquarium Herbarium ($n = 12$), University Herbarium at the University of California at Berkeley ($n = 8$), University of North Carolina at Chapel Hill ($n = 5$), San Diego Natural History Museum ($n = 4$), and University of Michigan ($n = 2$). These specimens included *Gelidium coulteri* ($n = 28$), *Gelidium robustum* ($n = 24$), *G. purpurascens* ($n = 13$), *Gelidium pusillum* ($n = 3$), *Gelidium arborescens* ($n = 1$) and one unidentified *Gelidium*. Archived *Gelidium* specimens were collected along a linear 6 km coastline from Point Pinos, Pacific Grove to Cannery Row, Monterey (electronic supplementary material, figure S1).

### (b) Curing experiment and historical reconstruction

To understand the use of herbaria specimens in reconstructing historical ocean conditions, we first examined the influence of preservation on tissue microchemistry. We sampled specimens at six regular intervals: 0.1, 1, 2, 4, 6 and 12 months after collection. Tissues were removed from each paper treatment in duplicate, avoiding epibionts, homogenized with a mortar and pestle, and transferred to cryovials (Thermo Scientific Nalgene, no. 5000-0020) or glass vials (Microsolv Technology Corporation, no. 9502S2CP) for stable isotope analyses. We sent prepared samples to Texas A&M University Protein Chemistry Laboratory for amino acid analyses, to California Animal Health & Food Safety Laboratory System for heavy metal screens, and Texas A&M University Stable Isotopes for Biosphere Science Laboratory for bulk stable isotope quantification (see the electronic supplementary material).

We fit loess (locally weighted regression, [24]) models to amino acid, heavy metal and total protein values over the 1 year curing period for all species combined for each paper type (electronic supplementary material, figures S2–S4). We calculated the per cent change between the loess value from the beginning to the end of the period. There was not enough tissue remaining at the 1 year mark to sample amino acid or total protein composition for one species-paper type treatment, or heavy metals for two species-paper type treatments. Additionally, we did not have replicate samples for amino acid and total protein analyses for two species-paper type or replicates for heavy metal analysis for one species-paper type treatment. We rescaled all values to account for these missing values and compare across species. To rescale, the values for each species were standardized by subtracting the species mean and dividing by the species s.d. We fitted loess models to stable isotope values over the 1 year curing period for all species (electronic supplementary material, figure S5) and combined by paper type. For these data, we did not rescale values as we had at least one sample from each species-paper type combination.

To examine the relationship between *Gelidium* microchemistry and environmental data, we obtained a variety of relevant data series. The traditional Bakun 3° monthly upwelling index [25], from the eastern North Pacific station (36° N 122° W) from 1946 to 2018, characterized coastal upwelling. Sardine catch records [26] throughout the cannery era, 1916–1967, reported trends in locally significant fisheries. Shore-based records (1919–2018, at Hopkins Marine Station) provided local sea surface temperatures, and the monthly Pacific Decadal Oscillation index

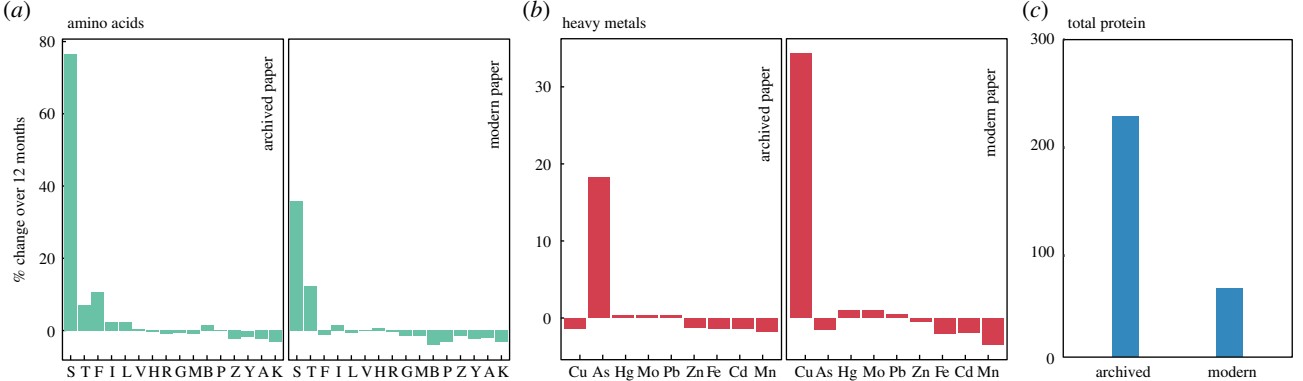

**Figure 1.** The composition of marine macroalgae changed over the 1 year curing process for several components but many exhibited little change. Per cent change over the 1 year period is shown by archived and modern paper for (*a*) amino acids, (*b*) heavy metals and (*c*) total protein. Amino acid units are in per cent by weight, heavy metals in ppm and total protein in per cent by weight. Values were rescaled by species. Amino acid abbreviations: serine (S), threonine (T), phenyl-alanine (F), isoleucine (I), leucine (L), valine (V), histidine (H), arginine (R), glycine (G), methionine (M), aspartic acid or asparagine (B), proline (P), glutamic acid or glutamine (Z), tyrosine (Y), alanine (A), lysine (K). Heavy metal is noted by standard elemental abbreviations. (Online version in colour.)

summarized ecosystem state [27]. The electronic supplementary material provides more details (electronic supplementary material, figures S6–S8) on these data series.

We corrected the seasonal sampling effects on stable isotope signatures by fitting a loess model to each stable isotope signature by Julian day. Each signature was rescaled by the ratio of the mean to the smoothed value for its Julian day. We fitted loess models to *Gelidium* spp. $\delta^{13}C$, $\delta^{15}N$ and $\delta^{18}O$, and upwelling index values over time. As we had no *a priori* expectations of appropriate loess spans, we fitted models using a range of spans (0.1–1, step 0.01). We plotted the upwelling index loess fits against each stable isotope loess fit to examine the linear regressions for each span (electronic supplementary material, figures S9–S11). The optimal span was the run with the lowest root mean square error (RMSE; electronic supplementary material, figure S12). As $\delta^{15}N$ most strongly correlated with upwelling, we determined the regression relationship of the two, relating the loess fits of upwelling index and $\delta^{15}N$ from the optimal span. With this formula, we hindcasted the expected upwelling index values that corresponded to the $\delta^{15}N$ recorded in historical herbaria samples.

## 3. Results

### (a) Curing effects

All but four amino acids maintained consistent composition in macroalgae over 1 year, not changing by more than a few percentage points (figure 1). Serine increased by the greatest percentage after 1 year in both archived paper (76.6%) and modern paper (35.8%). Threonine and phenylalanine changed greater than 10% on one paper type and less than 5% on the other. The remaining amino acids changed by less than 5% on both papers. *Ulva californica* had the highest variability of amino acid composition of all species (mean coefficient of variation (CV) = 10.8; Dataset S2), whereas *M. pyrifera* was the least variable (mean CV = 8.3; Dataset S2). Total protein (% by weight) increased by 227.0% on archived paper and increased by 64.5% on modern paper over 1 year (figure 1). Brown algae contained less total protein (*M. pyrifera:* mean = 10.4%, s.d. = 1.4; *S. compressa:* mean = 4.8%, s.d. = 0.6) than green or red algae species (see Dataset S2). *Cladophora columbiana* was the most variable in total protein over the year (CV = 17.9), *Cr. ruprechtiana* was the least variable (CV = 8.3).

Several heavy metals changed over the curing period, but most changes were negligible (figure 1). Cu increased by the greatest percentage across species on modern paper (34.3%) but decreased by 1.4% in archived paper. Interestingly, As increased 18.2% on archived paper but decreased 1.6% on modern paper. Cd, Fe, Pb, Mn, Hg, Mo and Zn all changed less than 4% on either paper. All species had low concentrations (less than 6 ppm) of Hg, Pb, Mo, Cd and Cu across the time period. Zn and Mn concentrations had higher concentrations but did not vary greatly between species (Zn: 14.8–35.5 ppm; Mn: 13.5–53.6 ppm; Dataset S2). Average As concentrations were highest in brown algae (74.5 ppm in *M. pyrifera* and 24.1 ppm in *S. compressa* versus 9.3–9.8 ppm in Chlorophyta and 4.9–5.6 ppm in Rhodophyta). Average Fe concentrations varied by species but showed no taxonomic pattern (highest average concentration: *Cl. columbiana* 1953.6 ppm; lowest: *G. purpurascens* 41.4 ppm). *Macrocystis pyrifera* had the most variable metal concentrations (mean CV: 95.9), whereas *Cl. columbiana* was the least variable (mean CV: 29.6).

We examined stable isotope ratios for all species by paper during the curing phase (electronic supplementary material, figure S5). For all species combined, $\delta^{13}C$ became enriched by 8.1% on archived paper and 3.6% on modern paper. $\delta^{15}N$ became enriched by 6.7% on archived paper and 7.2% on modern paper. $\delta^{18}O$ became depleted by 2.1% on archived paper and 0.3% on modern paper. Based on these results, we combined paper types to examine taxonomic trends. $\delta^{13}C$ did not change greatly over 1 year (Chlorophyta: +6.7%, Phaeophyceae: +4.2%, Rhodophyta +0.4%; figure 2). $\delta^{15}N$ increased most in Phaeophyceae (13.4%), followed by Rhodophyta (7.1%) and did not change for Chlorophyta (0.0%; figure 2). $\delta^{18}O$ increased most in Chlorophyta (6.9%), changed little for Rhodophyta (0.2%), and decreased in Phaeophyceae (−8.9%; figure 2).

By species, *M. pyrifera* was the most $\delta^{13}C$-enriched (mean = −12.6‰, s.d. = 0.9, range = −14.2 to −11.3; electronic supplementary material, figure S5, table S1). *Cryptopleura ruprechtiana* had far more depleted $\delta^{13}C$ values (mean = −30.0‰, SD = 1.0, range = −32.3 to −28.3) relative to the other species (electronic supplementary material, figure S5, table S1). *Macrocystis pyrifera* was the most $\delta^{15}N$-enriched (mean = 11.1‰, s.d. = 0.5, range = 11.1–13.0) of all species and *Gelidium purpurascens* most depleted (mean = 7.2‰, s.d. = 0.9, range = 5.5–8.6; electronic supplementary material, figure S5; table S1). *Gelidium purpurascens* was most $\delta^{18}O$-enriched (mean = 22.3‰, s.d. = 1.4, range = 19.3–23.7) and *Cr. ruprechtiana* was the most depleted (mean =

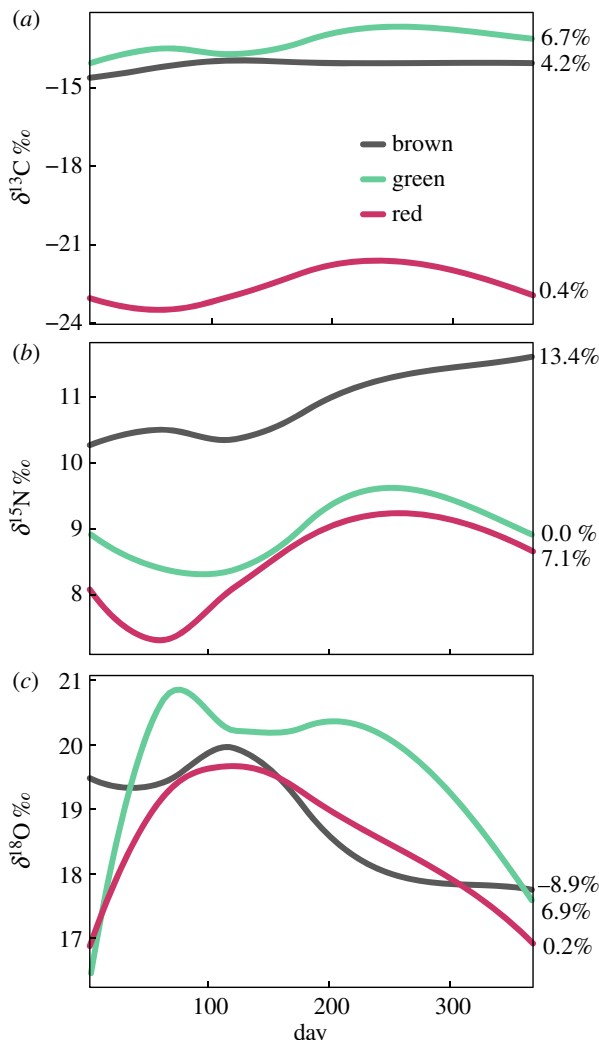

**Figure 2.** Stable isotope signatures of dried marine macroalgae changed the least in red algae species, followed by green algae and the most in brown algae after 1 year of curing. Loess curves were fit to stable isotopic signatures ((*a*) $\delta^{13}$C, (*b*) $\delta^{15}$N and (*c*) $\delta^{18}$O) for combined archived and modern paper types by algae taxonomic group. Green algae species included: *Cl. columbiana*, *U. californica*; red algae species included: *Cr. ruprechtiana*, *G. purpurascens*, brown algae species included: *M. pyrifera*, *S. compressa*. Loess curves with a span of 0.5 are shown for each taxonomic group across both paper types. Sampling occurred three days after collection, one month, two months, four months, six months and 1 year.

14.7‰, s.d. = 1.6, range = 12.2–16.9; electronic supplementary material, figure S5; table S1).

We performed all microchemistry analyses on the paper on which the macroalgae were pressed and dried. Archived paper from the 1930s was 1.1% total protein and modern herbarium paper purchased was 0.01% total protein. The amino acids with the greatest differences in composition by paper type were glycine (22.5% higher in archived paper), valine (6.8% lower in archived paper) and glutamic acid/glutamine (6.5% lower in archived paper; electronic supplementary material, table S2). Four heavy metals (Mn, Fe, Zn, Cu) were detected in higher concentrations in the archived paper than modern paper (electronic supplementary material, table S2). Pb was not detected in modern paper but was in archived paper (2.8 ppm). Hg, As, Mo and Cd were not detected in either paper. The $\delta^{13}$C value of archived paper was enriched (−23.8‰) relative to modern paper (−24.8‰). Nitrogen content was too low to determine $\delta^{15}$N

for either paper type. The $\delta^{18}$O value of archived paper was enriched (29.0‰) relative to modern paper (26.8‰).

## (b) Historical trends

The $\delta^{13}$C values in historical herbarium specimens of *Gelidium* spp. ranged from −30.9‰ (1878) to −11.0‰ (1970; $n = 70$, mean = −16.9‰, s.d. = 3.3). The $\delta^{13}$C values were most depleted from 1878, increased through to 1922 and declined from 1926 to 1949 (figure 3). No specimens were collected in the 1950s. The values then became enriched starting in 1963, peaked circa 1970 and became more depleted through 1986 (figure 3). From 1995 to 2018, *Gelidium* spp. $\delta^{13}$C values generally became more enriched (figure 3).

The $\delta^{15}$N values in historical samples ranged from 5.0‰ (1986) to 11.9‰ (1967; $n = 70$, mean = 8.5‰, s.d. = 1.5). $\delta^{15}$N values varied from 1878 through to early 1934. They became more depleted from 1937 to 1949, enriched in 1963–1981, then depleted through to 1986 and enriched from 1995 to 2018 (figure 3).

The $\delta^{18}$O values in historical samples ranged from 20.1‰ (1878) to 26.9‰ (2003; $n = 70$, mean = 22.8‰, s.d. = 1.6). $\delta^{18}$O values varied but became slightly more enriched from 1878 to a peak in 1945 followed by a decreasing trend to depleted values from 1963 to 1986 (figure 3). $\delta^{18}$O values became more enriched from 1986 to a peak in 2003 and have since become more depleted from 2010 to 2018 (figure 3).

## (c) Relationship to the environmental record

*Gelidium* spp. $\delta^{15}$N isotopic records, and $\delta^{13}$C to a lesser extent, correlated with trends in upwelling. The smoothed $\delta^{13}$C time series was positively correlated with smoothed upwelling index time series, with slope magnitude varying by loess span length (optimal span = 0.17, RMSE = 0.16; $y = -21.49 + 0.04x$, $F_{1,526} = 640$, adjusted $R^2 = 0.55$, $p < 0.001$; electronic supplementary material, figure S13). The slope direction of modelled $\delta^{18}$O versus upwelling varied by loess span length (optimal span = 0.72, RMSE = 0.23; $y = 23.89–0.01x$, $F_{1,526} = 86.02$, adjusted $R^2 = 0.14$, $p < 0.001$; electronic supplementary material, figure S13). The smoothed $\delta^{15}$N time series was positively correlated with the smoothed upwelling index time series, with slope magnitude varying by loess span length (optimal span = 0.72, RMSE = 0.05; $y = 5.75 + 0.03x$, $F_{1,526} = 7211$, adjusted $R^2 = 0.93$, $p < 0.001$; figure 4).

This strong relationship between $\delta^{15}$N and upwelling served to hindcast upwelling trends from 1878 using the historical herbaria isotope values preceding the Bakun upwelling index (figure 4). We used the above relationship ($y = 5.75 + 0.03x$) to calculate the upwelling index based on the *Gelidium* $\delta^{15}$N values prior to 1946. Because isotope values in red algae did not change by greater than 10% during the curing experiment or in a consistent direction, we can interpret the historical herbarium specimens without calibration. These hindcast upwelling trends record a decrease in upwelling mid-century (1937–1945; figure 4*c*).

## 4. Discussion

### (a) Curing process and macroalgae response

We observed variability in chemical signatures between species and within a single organism but did not observe consistent directional trends over the duration of the curing

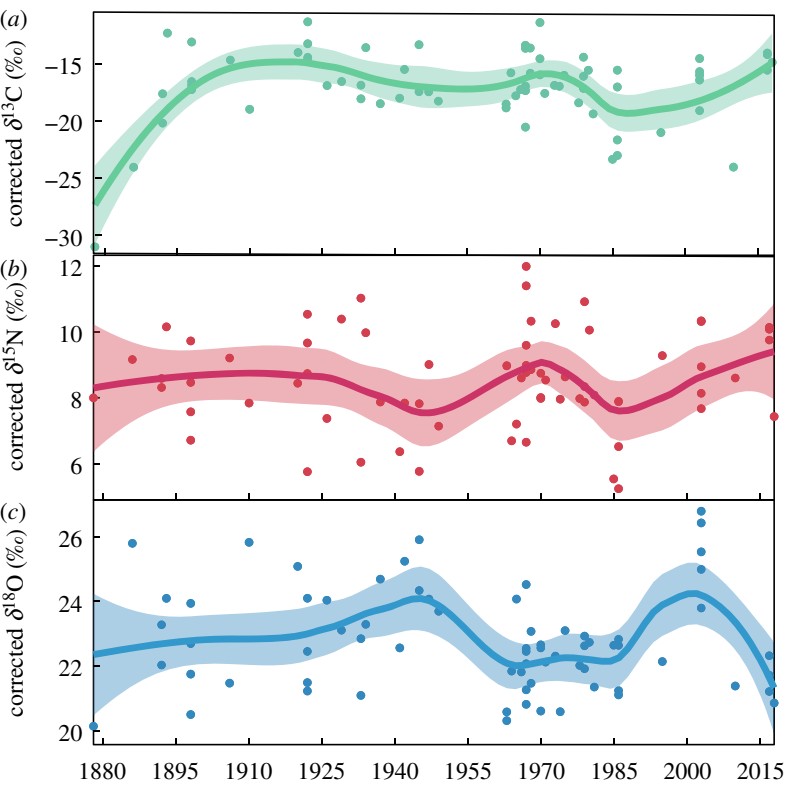

**Figure 3.** Isotopic signatures of historical archived specimens varied and changed throughout the 140 year time series. Signatures of (*a*) $\delta^{13}$C, (*b*) $\delta^{15}$N, and (*c*) $\delta^{18}$O were corrected for differences owing to seasonality by a rescaling ratio informed by the data. Data points each represent one *Gelidium* sp. specimen and smoothed loess curves are shown with loess confidence intervals. (Online version in colour.)

experiment. This is probably owing to differential uptake of nutrients and pollutants by individuals within a species based on their location within the intertidal, freshwater inputs, their exposure to wave action and resulting diffusion boundary layers, other nutrient limitations, light availability, temperature range and season [28,29]. We also expect differential uptake and storage within an individual based on the age of the tissue (actively growing apical fronds versus mature fronds), structure/function (blade versus stipe) and morphology as it relates to surface area to volume ratios [28,30,31]. In the curing experiment, we sampled across all parts of an individual to obtain sufficient material. This may account for some of the variability we observed within an organism. Total protein, for example, may be unevenly distributed throughout an individual organism's tissues. Among historical archived specimens, we sampled from distal branch tips so it is likely these values were less variable.

This variability highlights that species and tissues selected for study should match the scale of the research topic in question. Despite their variability, the results from our curing experiment may help inform this process. *Macrocystis pyrifera*, for example, had the most consistent amino acid composition over the curing process but was the most variable in heavy metal concentrations. *Macrocystis pyrifera* therefore may be reliable for amino acid inquiries but not for heavy metals. While consistency within the curing experiment is one metric in understanding species utility, overall per cent change from the time of collection until 1 year after pressing and drying may be most relevant in herbaria studies. Of the three taxonomic groups Rhodophyta demonstrated the lowest absolute change across all three stable isotope measures over 1 year (figure 2).

Species selection decisions should also consider structural integrity. Some species are known to degrade over time.

Carrageenan cell wall materials can break down allowing sulfuric acid to react with sugars and form a sticky black tar. We did not observe this process in any of our specimens over the year curing experiment. This degradation, however, has been observed in *Gigartina* and *Callophyllis* [32] and since reported in *Gloiocladia* and *Hymenocladia* by the Museum of New Zealand Te Papa Tongarewa.

We found differences between species consistent with known photosynthesis and growth patterns. Unlike other macroalgae, approximately 35% of red algae do not have carbon concentrating mechanisms (CCM) meaning they cannot concentrate bicarbonate and are reliant on $CO_2$ [33–35]. The low $\delta^{13}$C values of *Cr. ruprechtiana* indicate it also lacks a CCM, while all other species we examined use bicarbonate. This feature could make the species useful for answering $CO_2$-specific questions. $\delta^{15}$N ratios appear to be determined by growth rate and N availability, rather than by taxon. *Macrocystis pyrifera* consistently had the most enriched $\delta^{15}$N values relative to the other species while the other brown alga, *S. compressa*, was most depleted in $\delta^{15}$N. Tissue growth in *M. pyrifera* is rapid and predominantly annual [36] so it may be more N-limited than slower-growing, sympatric perennials. *Macrocystis pyrifera* may therefore be ideal for documenting short-term environmental dynamics. A macroalga that grows slowly, has high structural biomass, and a low N uptake rate would more slowly reflect the $\delta^{15}$N of its environment and should be considered for research questions reflecting larger scale, or longer term dynamics. Similarly, some macroalgae had elevated levels of certain heavy metals relative to the other species. *Cladophora columbiana* had high concentrations of Cu, Fe and Mn and *M. pyrifera* had high concentrations of As (electronic supplementary material, figure S3; table S1). *Cladophora columbiana* exhibits rapid opportunistic growth [37], while the cell wall chemistry

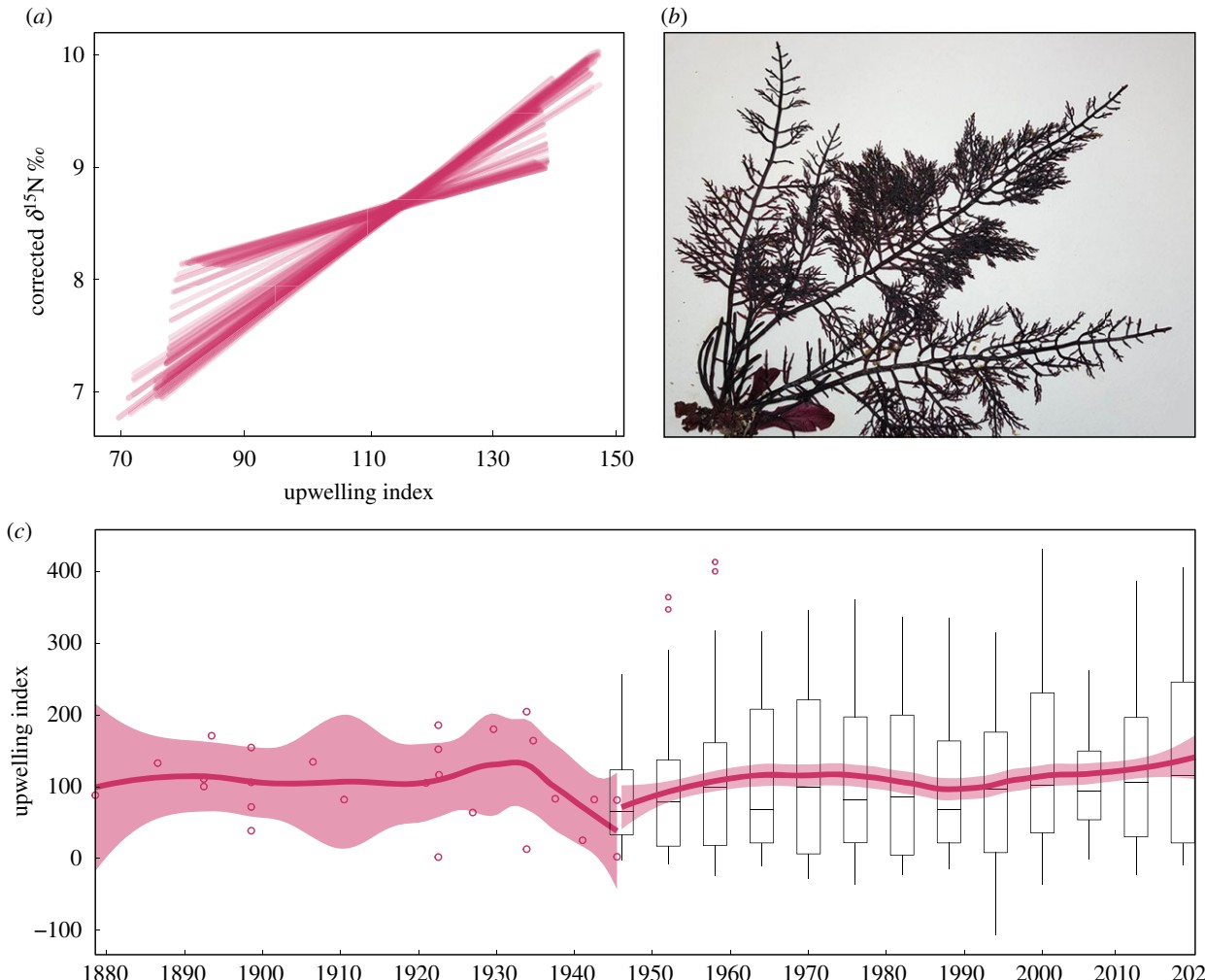

**Figure 4.** $\delta^{15}$N values of *Gelidium* spp. can hindcast historical upwelling trends prior to index estimates. (*a*) Loess models (spans = 0.1–1, interval = 0.01) were fitted to both time series of three-degree monthly Bakun upwelling index and seasonally corrected $\delta^{15}$N of herbarium *Gelidium* spp., 1946–2018. Linear regressions were fitted to these modelled $\delta^{15}$N values as a function of the modelled upwelling index values. Linear regressions for all model spans are shown here. (*b*) Image of *Gelidium* collected from Cabrillo Point, Pacific Grove, CA in 2017. (*c*) Monthly Bakun upwelling index values are shown in six-year binned boxplots, from 1946–2018, with loess model. Hindcast upwelling index values (open circles) were derived from $\delta^{15}$N values from historical herbaria *Gelidium* spp., 1878–1945, using the optimal linear regression relationship obtained from (*a*). A loess model is fitted to the hindcast data. (Online version in colour.)

of brown algae, like *M. pyrifera*, allows for high biosorption [38], making the group a potential heavy metal biomonitor [39]. These patterns reflect deeper Phylum level evolutionary traits, species-specific intrinsic biological processes, opportunistic life histories and surface area-to-volume ratios.

## (b) Historical record of upwelling

Regional productivity was correlated with $\delta^{15}$N from *Gelidium* herbaria specimens. Upwelling processes dominate the Monterey Bay nutrient regime [40]. The hindcast decrease in upwelling (1937–1945) supports the previously estimated long-term declines in upwelling and primary production from approximately 1920–1950 associated with the so-called 'sardine' regime [41,42]. These trends were associated with the peak in sardine catch through the 1930s and into the early 1940s followed by a crash from 1945 to 1950 as the system shifted to an upwelling-favourable, anchovy-dominant regime [41,42]. This crash resulted in the demise of the Monterey cannery industry.

Macroalgae $\delta^{15}$N trends track the upwelled N source in Monterey Bay. Growth rates in the perennial *Gelidium* spp. are highest after spring upwelling in summer or late summer

after frond regeneration begins following winter die back [43–47]. We expect the stable isotope values of the specimens, while integrated over the organism's lifespan, largely reflect this period. Inorganic N is available to macroalgae as ammonium excreted by invertebrates or nitrate brought to the surface by coastal upwelling [48]. Nitrate produces more enriched $\delta^{15}$N values than does ammonium so it follows that higher $\delta^{15}$N values in *Gelidium* spp. are correlated with higher upwelling index values [49]. Previous work has demonstrated that macroalgae fall into two functional groups, based on these N uptake pathways [48]. This complementarity reduces competition in diverse intertidal systems but is also a consideration when attempting to track specific environmental sources. Our results suggest *Gelidium* spp. prefer nitrate uptake. Additionally, the scale of the ammonium signal may act on a shorter timescale (e.g. tidally) than our data may detect.

The relatively high $\delta^{15}$N values we recorded probably reflect upwelling processes rather than wastewater inputs. Across studies, macroalgae in bays and estuaries with sewage and wastewater pollution have $\delta^{15}$N values of at least 6‰ or 8‰ [50–52]. All but one sample from either our curing experiment or in the historical herbaria had $\delta^{15}$N greater than 6.0‰, meeting this threshold. $\delta^{15}$N values of macroalgae do not correlate with

river discharge along the southern Central Coast [53]. There is not an available record of wastewater outflow volume or spills at this location, but this site in Monterey Bay has a long history of human settlement, with treated and untreated inputs probably present periodically throughout our time series. However, the $\delta^{15}$N of surface particulate available is naturally higher at middle and higher latitudes [54] and upwelling processes further increase $\delta^{15}$N values of primary producers [55].

The range of macroalgae $\delta^{15}$N values we observed is expected in an upwelling system. Nitrates entering the system are brought up from the deep waters during upwelling. While $\delta^{15}$N of nitrate in the central Pacific deep waters has been estimated at approximately 4.8–6‰ [54,56,57], it becomes enriched via strong denitrification by bacteria in the water column as these nutrient-rich but oxygen-poor waters are pushed to the surface [55]. The $\delta^{15}$N of the available nitrate pool becomes increasingly enriched as it reaches the surface to approximately 7‰ through fractionation associated with nitrate assimilation by phytoplankton [58] and 7.6‰ when nitrate is depleted [59]. Further, $\delta^{15}$N values of approximately 8‰ were recorded in upwelled nitrates along Monterey Bay from the northward transport of denitrified waters from the south [60]. Our similar values indicate that nitrate sources are being depleted during and following upwelling and that the $\delta^{15}$N of *Gelidium* spp. during its major period of growth is converging on the $\delta^{15}$N of the N source, a described phenomenon [54]. Isotopic discrimination decreases as the source pool is consumed and as growth rate increases [61]. *Gelidium* spp. therefore appear to be recording the upwelling and primary productivity processes of the larger Monterey Bay ecosystem.

## 5. Conclusion

Marine macroalgae specimens preserved in herbaria archives can provide a historical account of ocean conditions that precedes instrument records. The important Bakun upwelling index records for the California Current begin in 1946, and this timeline can be extended 70 years using isotopic analysis of *Gelidium* specimens. Our experimental results demonstrate that macroalgae pressed on paper and archived in herbaria are plentiful, accessible, inexpensively analysed and reliably interpreted. Researchers, however, should match specimens to their questions and metrics of interest. Our results here may specifically inform historical upwelling indices in other eastern boundary current systems that are similarly upwelling driven. Beyond this application, such analyses may also reveal pre-1950 nutrient baselines that may inform critical management issues in large-scale hypoxic zones [62] and coral reef ecosystems [14,63]. Additionally, owing to the limited availability of historical tissues, here we performed one longitudinal analysis using $\delta^{15}$N. Future studies that have access to more tissue samples may learn more from coupled approaches of two or more analyses [64]. For macroalgae, coupled analyses of amino acid content and bulk $\delta^{15}$N may inform both the total N flux as well as proximate source of ecosystem N [15].

Our results provide further justification for active, systematic collecting and archiving of marine macroalgae in herbaria. Higher resolution collections than we had available would allow examining of interacting environmental trends across multiple scales. Our time series from 1878 to 2018 has large gaps based on political and social factors—limited macroalgae collection during World War I, the depression, World War II and the 1950s, and the last four decades following the concerted effort by a few dedicated marine botanists [65]. Following this, specimen collection has largely been siloed for focused projects rather than comprehensive collecting across taxa, years and locations. Herbaria are critical to documenting environmental changes where instrumentation cannot, and recording how these changes directly affect living organisms, their shifting ranges and community interactions. Such macroalgae collections are underused and underfunded resources housed across the world's institutions. Maintaining herbaria is vital to advancing techniques to unlock long-term oceanographic data from archived macroalgal tissues and to developing management strategies to conserve coastal ecosystems.

Data accessibility. Datasets and analytical code for this study are included in third party repository at: https://osf.io/u7bxf/.

Authors' contributions. C.S. and K.V. designed the study. E.M., S.L. and K.V. collected and processed the specimens. E.M. performed the data analyses, and with K.V. generated the figures. E.M. wrote the manuscript with contributions from K.V. All authors reviewed the manuscript.

Competing interests. We declare we have no competing interests.

Funding. This research was supported by generous donor contributions to the Monterey Bay Aquarium.

Acknowledgements. In celebration of her 100th birthday, we dedicate this study to Isabella Kauakea Aiona Abbott, a Native Hawaiian and botanist, who exemplified traditions of her culture while making countless contributions to marine biology in the Pacific, who demonstrated that knowledge, humility, rigour and persistence can dwell together, and who championed the integral role and vital contributions of Indigenous peoples of the Pacific, Native Americans and women to science. J. Packard, J. Connor, R. Phillips, A. Bishop, A. Norton and L. San Ahmadi assisted with specimen collection and curation. H. Jariwala and L. San Ahmadi assisted with laboratory and specimen preparation. Several institutions generously contributed from their collections. These include the Gilbert H. Smith Collection at Stanford University (J. Thompson, M. Denny, and S. Palumbi), the Monterey Bay Aquarium Herbarium (R. Philips, J. Packard, J. Hoech, K. Regnier and K. Tuttle), the University of North Carolina Chapel Hill Herbarium (P. Gabrielson, C. A. McCormick), the University of Michigan Herbarium (B. Ruhfel), the San Diego Natural History Museum Herbarium (L. Hains, J. Rebman), the University and Jepson Herbarium at the University of California at Berkeley (K. Miller). K. Miller provided additional guidance through a macroalgae workshop that G. Wahlert facilitated. B. Dias, T. Farrugia, K. Tanaka, J. Fujii and T. Nicholson provided feedback on earlier versions of the manuscript.

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
