## [Reviewer comments · Proceedings of the Royal Society B: Biological Sciences]

Review History

RSPB-2020-0732.R0 (Original submission)

Review form: Reviewer 1

Recommendation

Major revision is needed (please make suggestions in comments)

Scientific importance: Is the manuscript an original and important contribution to its field?

Good

General interest: Is the paper of sufficient general interest?

Good

Quality of the paper: Is the overall quality of the paper suitable?

Marginal

Is the length of the paper justified?

Yes

Should the paper be seen by a specialist statistical reviewer?

Yes

Do you have any concerns about statistical analyses in this paper? If so, please specify them explicitly in your report.

No

It is a condition of publication that authors make their supporting data, code and materials available - either as supplementary material or hosted in an external repository. Please rate, if applicable, the supporting data on the following criteria.

Is it accessible?

Yes

Is it clear?

Yes

Is it adequate?

Yes

Do you have any ethical concerns with this paper?

No

Comments to the Author

The content of this paper would be worth for publication. The main story seems to be plausible and interesting to a wide readership. However, there are a number of points that need to be clarified before publication. The authors had better discuss about the possibilities of $\delta^{15}\text{N}$ of *Gelidium* spp. specimen as a proxy for century-scale upwelling trends in Central California. These are given below.

L. 106-107: Was the influence of preservation on C and N contents of macroalgal tissue examined?

L. 121-122: How did the authors specifically rescaled? How should Figure 1 and Figure S2-4 be compared?

L. 125-126: Did the authors examine the relationship between amino acids, total protein, heavy metals (except for $\delta^{13}\text{C}$, $\delta^{15}\text{N}$, $\delta^{18}\text{O}$) of *Gelidium* spp. and a variety of relevant environmental data series?

L. 130-131: In this study, the relationship between one isotopic record and environmental data was examined (e.g., the $\delta^{15}\text{N}$ and sardine catch: Figure S6, the $\delta^{18}\text{O}$ and sea surface temperature: Figure S7). Was the relationship between one isotopic record and multiple environmental data examined? Trends in upwelling were correlated with *Gelidium* spp. $\delta^{15}\text{N}$ records. Upwelling is also expected to be lower sea surface temperature. Were sea surface temperature data correlated with upwelling index, or with *Gelidium* spp. $\delta^{15}\text{N}$ records?

L. 150-152, Figure 1: Why did total protein increase?

L. 190, Table S2: Nitrogen content was too low to determine $\delta^{15}\text{N}$ for either paper type, so the $\delta^{15}\text{N}$ values should not be described.

L. 215-216, Figure 4c: The authors had better discuss about the possibilities of $\delta^{15}\text{N}$ of *Gelidium* spp. specimen as a proxy for century-scale upwelling trends in Central California. Were hindcast upwelling index values derived from $\delta^{15}\text{N}$ values historical herbaria *Gelidium* spp., 1878-1945 validated using the other environmental data? $\delta^{15}\text{N}$ analysis of sediment cores in this region would be also helpful for the discussion.

L. 232-233, L. 275-277, L.307-308: In spite of the perennial *Gelidium* spp., did the distal branch tips record the upwelling and primary production of the collection time? When did the distal branch

tips of *Gelidium* spp. record ocean conditions?

L. 248-266: The authors should describe the influence of preservation on tissue microchemistry to understand the utility of herbaria specimens in reconstructing historical ocean conditions. However, was it necessary to describe L.248-266?

L. 287-288: Did the high $\delta^{15}\text{N}$ values likely reflect upwelling processes rather than wastewater inputs? How far was macroalgae collected from the shore? Did the authors examine the relationship between the population and the $\delta^{15}\text{N}$ records? Moreover, were several heavy metals the indicator of wastewater inputs?

L. 300-303: $\delta^{15}\text{N}$ records *Gelidium* spp. specimen were correlated with trends in upwelling. However, the $\delta^{15}\text{N}$ values of *Gelidium* spp. specimen were in the range of 8-10 ‰, which were higher than that of ~8 ‰ recorded in upwelled nitrates along Monterey Bay from the northward transport of denitrified waters from the south (especially around 1970 and after 2000). Why were the $\delta^{15}\text{N}$ values of *Gelidium* spp. specimen higher?

L. 302-303: Did the authors obtain relevant data series to describe the temporal variation of NO_2 , NO_3 , and NH_4 concentrations in this region?

L. 307-308: Did the authors obtain relevant data series to describe the temporal variation of the biomass of sea algae or phytoplankton (chlorophyll-a concentration) in this region? If obtained, did the authors examine the relationship between the upwelling index and the primary production data? When the $\delta^{15}\text{N}$ values of *Gelidium* spp. specimen were high around 1970 and after 2000, were the upwelling index and primary production high? Similarly, when the $\delta^{15}\text{N}$ values were low around 1985, were the upwelling index and primary productivity low?

There are some typographical errors.

L. 150, 153, 161: Are they "Table S1"?

L. 170-174: Are they not "‰" but "%"?

L. 176-181: Are they not "Table S2" but "Table S1"?

L. 186-187: Are they not "Table S1" but "Table S2"?

L. 209: Is it not "0.17" but "0.71"?

L. 317: Is it not "than" but "that"?

#: As a technical comment, "delta" in $\delta^{13}\text{C}$, $\delta^{15}\text{N}$, and $\delta^{18}\text{O}$ should be italicized.

Review form: Reviewer 2

Recommendation

Accept with minor revision (please list in comments)

Scientific importance: Is the manuscript an original and important contribution to its field?

Good

General interest: Is the paper of sufficient general interest?

Acceptable

Quality of the paper: Is the overall quality of the paper suitable?

Excellent

Is the length of the paper justified?

Yes

Should the paper be seen by a specialist statistical reviewer?

No

Do you have any concerns about statistical analyses in this paper? If so, please specify them explicitly in your report.

No

It is a condition of publication that authors make their supporting data, code and materials available - either as supplementary material or hosted in an external repository. Please rate, if applicable, the supporting data on the following criteria.

Is it accessible?

Yes

Is it clear?

Yes

Is it adequate?

Yes

Do you have any ethical concerns with this paper?

No

Comments to the Author

This is a careful study. I'm especially impressed with how the paper and algae were studied to determine their utility in this study. It could have applications in other places but it requires long-term herbaria specimens. The minor revision I'd like to see is to be clear why this is important and how it can be applied to other settings.

Decision letter (RSPB-2020-0732.R0)

27-Apr-2020

Dear Dr Miller:

Your manuscript has now been peer reviewed and the reviews have been assessed by an Associate Editor. The reviewers' comments (not including confidential comments to the Editor) and the comments from the Associate Editor are included at the end of this email for your reference. As you will see, the reviewers and the Editors have raised some concerns with your manuscript and we would like to invite you to revise your manuscript to address them.

Research ethics:

Use of animals and field studies:

Please submit a copy of your revised paper within three weeks. If we do not hear from you within this time your manuscript will be rejected. If you are unable to meet this deadline please let us know as soon as possible, as we may be able to grant a short extension.

Best wishes,
Dr Daniel Costa
mailto:proceedingsb@royalsociety.org

Associate Editor
Board Member: 1
Comments to Author:
24 April 2020

Dear Dr. Miller,

Thank you for submitting your manuscript entitled “Herbaria macroalgae as a proxy for century-scale upwelling trends in Central California” for consideration as a research paper in PRSB. I have received two reviews of your submission and have read your paper at length: this is an interesting paper that has potential to be published in PRSB.

Both reviewers are supportive of your submission, with one being very brief, and the other offering a number of self-explanatory suggestions. While I am disappointed at the brevity of one of the reviews, it does come from a phycologist of international stature who is in a good position to judge the quality of your submission. Their support is valuable.

The appeal of your work focuses on the potential benefit that can be gleaned from the information stored in archived algal samples at a time when there is acute interest in reconstructing the past. However, your work balances the presentation of a method with its application through hind-casting upwelling trends, and a methods paper is not a strong candidate for PRSB. This limitation is countered by the potentially large and positive impacts of this method to the field, and the results present from its sample application. The key issue is described in the brief review: “The minor revision I'd like to see is to be clear why this is important and how it can be applied to other settings.” The appeal of your submission to PRSB would be enhanced by revising your text to emphasize the importance of the method through the answers it can deliver to important and long-standing questions.

Thank you for submitting your manuscript to PRSB

Yours sincerely,
Peter J. Edmunds PhD
Board Member

Reviewer(s)' Comments to Author:

Referee: 1

Comments to the Author(s)

The content of this paper would be worth for publication. The main story seems to be plausible and interesting to a wide readership. However, there are a number of points that need to be clarified before publication. The authors had better discuss about the possibilities of $\delta 15\text{N}$ of *Gelidium* spp. specimen as a proxy for century-scale upwelling trends in Central California. These are given below.

L. 106-107: Was the influence of preservation on C and N contents of macroalgal tissue examined?

L. 121-122: How did the authors specifically rescaled? How should Figure 1 and Figure S2-4 be compared?

L. 125-126: Did the authors examine the relationship between amino acids, total protein, heavy metals (except for $\delta 13\text{C}$, $\delta 15\text{N}$, $\delta 18\text{O}$) of *Gelidium* spp. and a variety of relevant environmental data series?

L. 130-131: In this study, the relationship between one isotopic record and environmental data was examined (e.g., the $\delta 15\text{N}$ and sardine catch: Figure S6, the $\delta 18\text{O}$ and sea surface temperature: Figure S7). Was the relationship between one isotopic record and multiple environmental data examined? Trends in upwelling were correlated with *Gelidium* spp. $\delta 15\text{N}$ records. Upwelling is also expected to be lower sea surface temperature. Were sea surface temperature data correlated with upwelling index, or with *Gelidium* spp. $\delta 15\text{N}$ records?

L. 150-152, Figure 1: Why did total protein increase?

L. 190, Table S2: Nitrogen content was too low to determine $\delta 15\text{N}$ for either paper type, so the $\delta 15\text{N}$ values should not be described.

L. 215-216, Figure 4c: The authors had better discuss about the possibilities of $\delta 15\text{N}$ of *Gelidium* spp. specimen as a proxy for century-scale upwelling trends in Central California. Were hindcast upwelling index values derived from $\delta 15\text{N}$ values historical herbaria *Gelidium* spp., 1878-1945 validated using the other environmental data? $\delta 15\text{N}$ analysis of sediment cores in this region would be also helpful for the discussion.

L. 232-233, L. 275-277, L.307-308: In spite of the perennial *Gelidium* spp., did the distal branch tips record the upwelling and primary production of the collection time? When did the distal branch tips of *Gelidium* spp. record ocean conditions?

L. 248-266: The authors should describe the influence of preservation on tissue microchemistry to understand the utility of herbaria specimens in reconstructing historical ocean conditions. However, was it necessary to describe L.248-266?

L. 287-288: Did the high $\delta 15\text{N}$ values likely reflect upwelling processes rather than wastewater inputs? How far was macroalgae collected from the shore? Did the authors examine the relationship between the population and the $\delta 15\text{N}$ records? Moreover, were several heavy metals the indicator of wastewater inputs?

L. 300-303: $\delta 15\text{N}$ records *Gelidium* spp. specimen were correlated with trends in upwelling. However, the $\delta 15\text{N}$ values of *Gelidium* spp. specimen were in the range of 8-10 ‰, which were higher than that of ~8 ‰ recorded in upwelled nitrates along Monterey Bay from the northward

transport of denitrified waters from the south (especially around 1970 and after 2000). Why were the $\delta^{15}\text{N}$ values of *Gelidium* spp. specimen higher?

L. 302-303: Did the authors obtain relevant data series to describe the temporal variation of NO_2 , NO_3 , and NH_4 concentrations in this region?

L. 307-308: Did the authors obtain relevant data series to describe the temporal variation of the biomass of sea algae or phytoplankton (chlorophyll-a concentration) in this region? If obtained, did the authors examine the relationship between the upwelling index and the primary production data? When the $\delta^{15}\text{N}$ values of *Gelidium* spp. specimen were high around 1970 and after 2000, were the upwelling index and primary production high? Similarly, when the $\delta^{15}\text{N}$ values were low around 1985, were the upwelling index and primary productivity low?

There are some typographical errors.

L. 150, 153, 161: Are they "Table S1"?

L. 170-174: Are they not "%" but "%o"?

L. 176-181: Are they not "Table S2" but "Table S1"?

L. 186-187: Are they not "Table S1" but "Table S2"?

L. 209: Is it not "0.17" but "0.71"?

L. 317: Is it not "than" but "that"?

#: As a technical comment, "delta" in $\delta^{13}\text{C}$, $\delta^{15}\text{N}$, and $\delta^{18}\text{O}$ should be italicized.

Referee: 2

Comments to the Author(s)

This is a careful study. I'm especially impressed with how the paper and algae were studied to determine their utility in this study. It could have applications in other places but it requires long-term herbaria specimens. The minor revision I'd like to see is to be clear why this is important and how it can be applied to other settings.

Author's Response to Decision Letter for (RSPB-2020-0732.R0)

See Appendix A.

Decision letter (RSPB-2020-0732.R1)

24-May-2020

Dear Dr Miller

I am pleased to inform you that your manuscript entitled "Herbaria macroalgae as a proxy for historical upwelling trends in Central California" has been accepted for publication in Proceedings B.

Open Access

Paper charges

Sincerely,

Dr Daniel Costa

Associate Editor:

Board Member

Comments to Author:

12 May 2020

Dear Dr. Miller,

Thank you for returning your revised submitting your manuscript entitled "Herbaria macroalgae as a proxy for historical upwelling trends in Central California" for further consideration as a research paper in PRSB. I have read your revision and refreshed my memory regarding your submission history (and initial reviews) and am happy with the changes you have made. I think your submission can make a very nice contribution to PRSB.

Yours sincerely,

Peter J. Edmunds PhD

Board Member

Appendix A

Editor Comments to Author:

The appeal of your work focuses on the potential benefit that can be gleaned from the information stored in archived algal samples at a time when there is acute interest in reconstructing the past. However, your work balances the presentation of a method with its application through hind-casting upwelling trends, and a methods paper is not a strong candidate for PRSB. This limitation is countered by the potentially large and positive impacts of this method to the field, and the results present from its sample application. The key issue is described in the brief review: “The minor revision I'd like to see is to be clear why this is important and how it can be applied to other settings.” The appeal of your submission to PRSB would be enhanced by revising your text to emphasize the importance of the method through the answers it can deliver to important and long-standing questions.

Thank you, we agree that the method and case-study presented in this paper provide important and wider applications that need to be better highlighted. We have clarified the importance of this study and its applications to other systems and questions in the Abstract (lines 24-26, 34-36), Introduction (lines 74-76) and the Discussion (lines 323-331).

Reviewer(s)' Comments to Author:

Referee: 1

Comments to the Author(s)

The content of this paper would be worth for publication. The main story seems to be plausible and interesting to a wide readership. However, there are a number of points that need to be clarified before publication. The authors had better discuss about the possibilities of $\delta^{15}\text{N}$ of *Gelidium* spp. specimen as a proxy for century-scale upwelling trends in Central California.

These are given below.

Thank you for these general and specific comments. They greatly improved the clarity of the manuscript.

L. 106-107: Was the influence of preservation on C and N contents of macroalgal tissue examined?

These data will be available with the online material. No consistent trends were observed and due to space restraints we did not describe every parameter within the text.

L. 121-122: How did the authors specifically rescaled? How should Figure 1 and Figure S2-4 be compared?

We have revised the methods section detailing the rescaling steps in lines 124-125 for clarity. Figure 1 and Figures S2-4 directly comparable. Figure 1 is a summary figure of the data in Figures S2-4. We calculated the percent difference between loess value at one year and the

loess value at the beginning from Figures S2-4 and those percent change values are plotted in Figure 1 (described in lines 117-119).

L. 125-126: Did the authors examine the relationship between amino acids, total protein, heavy metals (except for $\delta^{13}\text{C}$, $\delta^{15}\text{N}$, $\delta^{18}\text{O}$) of *Gelidium* spp. and a variety of relevant environmental data series?

We selected one set of analyses (stable isotope analyses) to examine in our case study because we wanted to avoid taking repeated samples from priceless specimens. We chose stable isotope analyses because we knew these would be interesting metrics for our ecosystem and because red algae were relatively consistent for these analyses in the curing experiment.

L. 130-131: In this study, the relationship between one isotopic record and environmental data was examined (e.g., the $\delta^{15}\text{N}$ and sardine catch: Figure S6, the $\delta^{18}\text{O}$ and sea surface temperature: Figure S7). Was the relationship between one isotopic record and multiple environmental data examined? Trends in upwelling were correlated with *Gelidium* spp. $\delta^{15}\text{N}$ records. Upwelling is also expected to be lower sea surface temperature. Were sea surface temperature data correlated with upwelling index, or with *Gelidium* spp. $\delta^{15}\text{N}$ records?

One isotopic record was examined against one environmental dataset. Sea surface temperatures did not correlate with $\delta^{15}\text{N}$ records. We used multiple smoothing spans but did not find a correlation. The SST records are highly variable across days while upwelling has a long-term effect so that is possibly why the $\delta^{15}\text{N}$ was correlated with upwelling and not temperature. Because we did not observe the individual correlation, we did not feel the multiple correlation would enhance the results.

L. 150-152, Figure 1: Why did total protein increase?

Protein may be unevenly distributed throughout an individual organism's tissues. This explanation has been added to the discussion text at lines 239-241.

L. 190, Table S2: Nitrogen content was too low to determine $\delta^{15}\text{N}$ for either paper type, so the $\delta^{15}\text{N}$ values should not be described.

Revised as requested in Table S2.

L. 215-216, Figure 4c: The authors had better discuss about the possibilities of $\delta^{15}\text{N}$ of *Gelidium* spp. specimen as a proxy for century-scale upwelling trends in Central California. Were hindcast upwelling index values derived from $\delta^{15}\text{N}$ values historical herbaria *Gelidium* spp., 1878-1945 validated using the other environmental data? $\delta^{15}\text{N}$ analysis of sediment cores in this region would be also helpful for the discussion.

Revised as requested (lines 221-223).

L. 232-233, L. 275-277, L.307-308: In spite of the perennial *Gelidium* spp., did the distal branch tips record the upwelling and primary production of the collection time? When did the distal branch tips of *Gelidium* spp. record ocean conditions?

This is a good question and one that is not entirely known in published literature. As mentioned in lines 282-28 we expect the distal branches to reflect the growth following spring upwelling because this is the main period of growth (1-5). So, depending on collection time that could be relatively recent or reflect the previous year. For this reason, we tested multiple smoothing span lengths in the loess models to examine the correlations over multiple time scales.

1. Carter A.R., Anderson R.J. 1986 Seasonal growth and agar contents in *Gelidium pristoides* (Gelidiales, Rhodophyta) from Port Alfred, South Africa. *Botanica Marina* **29**, 117-123.
2. Gorostiaga J.M. 1994 Growth and production of the red alga *Gelidium sesquipedale* off the Basque coast (northern Spain). *Marine Biology* **120**, 311-322.
3. Santos R. 1994 Frond dynamics of the commercial seaweed *Gelidium sesquipedale*: effects of size and of frond history. *Marine Ecology Progress Series* **107**, 295-305.
4. Rico J.M. 1991 Field studies and growth experiments on *Gelidium-Latifolium* from Asturias (Northern Spain). *Hydrobiologia* **221**, 67-5.
5. Freile-Pelegrin Y., Robledo D., Serviere-Zaragoza E. 1999 *Gelidium robustum* agar: quality characteristics from exploited beds and seasonality from an unexploited bed at Southern Baja California, Mexico. *Hydrobiologia* **398/399**, 501-507.

L. 248-266: The authors should describe the influence of preservation on tissue microchemistry to understand the utility of herbaria specimens in reconstructing historical ocean conditions. However, was it necessary to describe L.248-266?

Revised to shorten as requested (by ~30 words) and include less background. See lines 257-259. Taxa-specific response of macroalgae to preservation is one of the main findings of the curing experiment so it requires some explanation in the discussion to inform researchers who may select species based on our results.

L. 287-288: Did the high $\delta^{15}\text{N}$ values likely reflect upwelling processes rather than wastewater inputs? How far was macroalgae collected from the shore? Did the authors examine the relationship between the population and the $\delta^{15}\text{N}$ records? Moreover, were several heavy metals the indicator of wastewater inputs?

Yes, the $\delta^{15}\text{N}$ values are expected in an upwelling system such as this. The values are consistently in this range from 1878-2018, a period with dramatically increasing population. We looked for population records, but they were patchy for the early part of the time series. $\delta^{15}\text{N}$ values at the beginning and end of the time series did not differ greatly. The trend was far from an exponential slope as would have been the case from the population for the first half of the time series. Macroalgae was collected from the intertidal and subtidal at a marine protected area where formerly, commercial vessels were repaired. Many metals are selectively absorbed in brown algae (1) and can be found in kelps in relatively pristine habitats so we cannot directly correlate heavy metals with wastewater.

1. Hashim M.A., Chu K.H. 2004 Biosorption of cadmium by brown, green, and red seaweeds. *Chemical Engineering Journal* **97**, 249-255.

L. 300-303: $\delta^{15}\text{N}$ records *Gelidium* spp. specimen were correlated with trends in upwelling. However, the $\delta^{15}\text{N}$ values of *Gelidium* spp. specimen were in the range of 8-10 ‰, which were higher than that of ~8 ‰ recorded in upwelled nitrates along Monterey Bay from the northward transport of denitrified waters from the south (especially around 1970 and after 2000). Why were the $\delta^{15}\text{N}$ values of *Gelidium* spp. specimen higher?

The high $\delta^{15}\text{N}$ values are reflective of rapid growth and depleted nitrate pool. Phytoplankton assimilate $\delta^{15}\text{N}$ more rapidly than macroalgae leaving a more depleted nitrate ($\delta^{15}\text{N}$ enriched)

pool.

L. 302-303: Did the authors obtain relevant data series to describe the temporal variation of NO₂, NO₃, and NH₄ concentrations in this region?

No, unfortunately the data series did not have the temporal resolution necessary to describe these nitrogenous compound dynamics.

L. 307-308: Did the authors obtain relevant data series to describe the temporal variation of the biomass of sea algae or phytoplankton (chlorophyll-a concentration) in this region? If obtained, did the authors examine the relationship between the upwelling index and the primary production data? When the $\delta^{15}\text{N}$ values of *Gelidium* spp. specimen were high around 1970 and after 2000, were the upwelling index and primary production high? Similarly, when the $\delta^{15}\text{N}$ values were low around 1985, were the upwelling index and primary productivity low?

We did not use primary production data because the records did not extend far enough historically to correlate with our dataset. Yes, the upwelling index followed the trends the reviewer described.

There are some typographical errors.

L. 150, 153, 161: Are they "Table S1"?

Thank you, corrected as suggested. We moved data from Table S1 to Dataset S2 due to length.

L. 170-174: Are they not "%" but "%o"?

Thank you, corrected as suggested.

L. 176-181: Are they not "Table S2" but "Table S1"?

Thank you, corrected as suggested.

L. 186-187: Are they not "Table S1" but "Table S2"?

Thank you, corrected as suggested.

L. 209: Is it not "0.17" but "0.71"?

No, it should remain as 0.17.

L. 317: Is it not "than" but "that"?

We have reworded for clarity. It should remain "than" but we understand it was not clear and have revised.

#: As a technical comment, "delta" in $\delta^{13}\text{C}$, $\delta^{15}\text{N}$, and $\delta^{18}\text{O}$ should be italicized.

Thank you, corrected throughout.

Referee: 2

Comments to the Author(s)

This is a careful study. I'm especially impressed with how the paper and algae were studied to determine their utility in this study. It could have applications in other places but it requires long-term herbaria specimens. The minor revision I'd like to see is to be clear why this is important and how it can be applied to other settings.

Thank you, we clarified the importance of this study and how it can be applied in other systems in the Introduction (lines 74-76) and the Discussion (lines 323-331). We also revised the Abstract to better reflect these revisions.